# Efficient Systolic Array Based on Decomposable MAC for Quantized Deep Neural Networks

## Abstract

Deep Neural Networks (DNNs) have achieved high accuracy in various machine learning applications in recent years. As the recognition accuracy of deep learning applications increases, reducing the complexity of these neural networks and performing the DNN computation on embedded systems or mobile devices become an emerging and crucial challenge. Quantization has been presented to reduce the utilization of computational resources by compressing the input data and weights from floating-point numbers to integers with shorter bit-width. For practical power reduction, it is necessary to operate these DNNs with quantized parameters on appropriate hardware. Therefore, systolic arrays are adopted to be the major computation units for matrix multiplication in DNN accelerators. To obtain a better tradeoff between the precision/accuracy and power consumption, using parameters with various bit-widths among different layers within a DNN is an advanced quantization method. In this paper, we propose a novel decomposition strategy to construct a low-power decomposable multiplier-accumulator (MAC) for the energy efficiency of quantized DNNs. In the experiments, when 65% multiplication operations of VGG-16 are operated in shorter bit-width with at most 1% accuracy loss on the CIFAR-10 dataset, our decomposable MAC has 50% energy reduction compared with a non-decomposable MAC.

## 1 Introduction

In the past decade, deep neural networks (DNNs) have achieved high accuracy in various machine learning applications, such as image classification, object detection, and speech recognition. AlexNet (Krizhevsky et al., 2012) applied 8 layers and 60 million parameters to achieve a top-5 error rate of 15.3%, which won the championship in the ILSVRC2012 competition, and thus has driven research trends on DNNs and advanced machine learning applications. In the following years, Szegedy et al. (2015); Simonyan & Zisserman (2014); He et al. (2016); Huang et al. (2017) presented to apply deeper neural networks with more layers and parameters for higher accuracy.

As the accuracy of deep learning applications increases, optimizing these complex networks to be simpler, more computational-efficient, or even suitable for embedded systems and mobile devices becomes an emerging and crucial challenge. Therefore, many methods have been proposed to reduce the complexity of neural networks. For example, pruning redundant connections within the model (Han et al., 2015; Zhu & Gupta, 2017), reducing the kernels/filters within layers (Liu et al., 2017; Luo et al., 2017; Howard et al., 2017), removing several layers (Wen et al., 2016; Li et al., 2016), or using fewer bits (also known as quantization) during the computation (Lin et al., 2016; Zhou et al., 2017; Jouppi et al., 2017; Hubara et al., 2017; Gysel et al., 2018; Mishra et al., 2017; Jacob et al., 2018) have been presented to streamline DNNs with negligible accuracy loss.

According to the accuracy requirements of an application and its associated DNN structure, the bit-width of the setting of parameters for the DNN may be different from one layer to another. In addition, the activations and weights in some DNNs can be squeezed into 8 bits or less without significant accuracy loss (Gysel et al., 2018; Mishra et al., 2017). Fig. 1a reveals the accuracy (normalized to that of the original DNN operated with 32-bit floating-point activations and weights) on CIFAR-10 dataset of DNNs with the activations and weights being quantized into uniform bit-

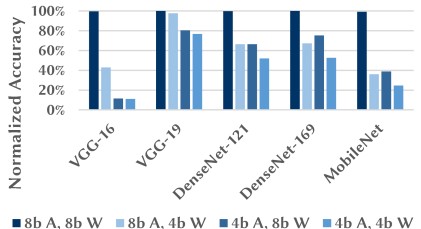
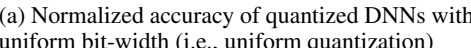
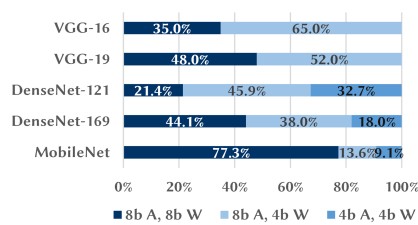

(a) Normalized accuracy of quantized DNNs with uniform bit-width (i.e., uniform quantization)

(b) Breakdown of layer-wise quantization with at most 1% accuracy loss

Figure 1: Tradeoff between accuracy and bit-width of quantized activations (A) and weights (W)

width, 8 bits or 4 bits, across all layers within the specific neural networks. The accuracy loss can be less than 1% when 8-bit activations and 8-bit weights are utilized in these DNNs. However, directly compressing all activations and/or weights into 4 bits might devastate the accuracy. Instead of quantizing all the parameters within a neural network to uniform bit-width, adjusting the bit-width of quantized parameters from layer to layer (i.e., layer-wise quantization) can obtain a better tradeoff between required computing resources and resulting accuracy (Mishra et al., 2017; Jacob et al., 2018). Based on the quantization schemes presented by Lin et al. (2016); Jacob et al. (2018), Fig. 1b shows the breakdown of layer-wise quantization, in terms of the percentage of multiplication-accumulations operations, within the quantized DNNs by trading with at most 1% accuracy loss.

For practical reduction in power consumption, it is necessary to run these DNNs with quantized parameters on appropriate hardware. Fig. 2a illustrates a simple DNN with multiple hidden layers. Within each layer, activations from the previous layer are multiplied by kernels/filters to obtain the outputs of the current layer; throughout all layer in a DNN, numerous multiplications and accumulations are involved. To accelerate DNN operations, Jouppi et al. (2017); Zhang et al. (2018); Gupta et al. (2015); Chen et al. (2017) employ a systolic array of multiplier-accumulators (MACs) (see Fig. 2b) as their major computation units for matrix multiplication, which perform 8-bit integer multiplication and 16-bit fixed-point multiplication, respectively.

To obtain a better tradeoff between accuracy and power consumption by quantizing parameters with various bit-width among different layers (i.e., layer-wise quantization), the processing elements for DNNs should support multiplication operations with variable bit-widths. Configurable multipliers (Sharma et al., 2018; Pfänder et al., 2004; Bermak et al., 1997; Haynes & Cheung, 1998) are presented to adjust the bit-width of operating MACs for power saving, when shorter bit-width is sufficient to satisfy the accuracy requirement; when longer bit-width is used, they generate the product of multiplication by combining the outputs from neighbor small MACs. However, the power/energy overhead for the combination of outputs from MAC neighbors may be significant and should not be ignored. In other words, layer-wise quantization needs to supported by customized hardware; otherwise the benefit in terms of power/energy consumption will be slashed (cut down significantly).

In this paper, we propose a novel decomposition strategy for low-power decomposable multiplication-accumulation. Based on the strategy, multiplication with longer bit-width can be departed and composed by its partial products with less computational quantity. In addition, accord-

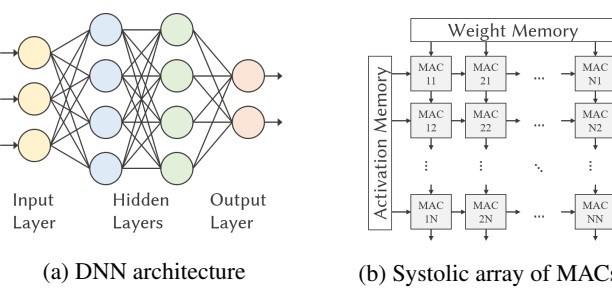

(a) DNN architecture

(b) Systolic array of MACs

Figure 2: DNN and systolic array

ing to the bit-width requirement of the current computation, a decomposable MAC based on our strategy can dynamically adjust the bit-width of its final multiplication result. In addition, according to the bit-width requirement of the current computation, a decomposable MAC based on our strategy can dynamically adjust the bit-width of its final multiplication result. Instead of generating multiplication results with required bit-width directly by connecting signals among neighbor MACs for composition as straightforward/intuitive configurable MACs (Sharma et al., 2018; Pfänder et al., 2004; Bermak et al., 1997; Haynes & Cheung, 1998), our strategy reduces the selection signals for the operation bit-width of multiplication units. Therefore, our strategy can obtain significant energy efficiency from multiplication with variable bit-widths.

## 2 BASIC IDEA OF OUR PROPOSED DECOMPOSITION STRATEGY

### 2.1 EXAMPLE OF OUR DECOMPOSABLE MAC OPERATION

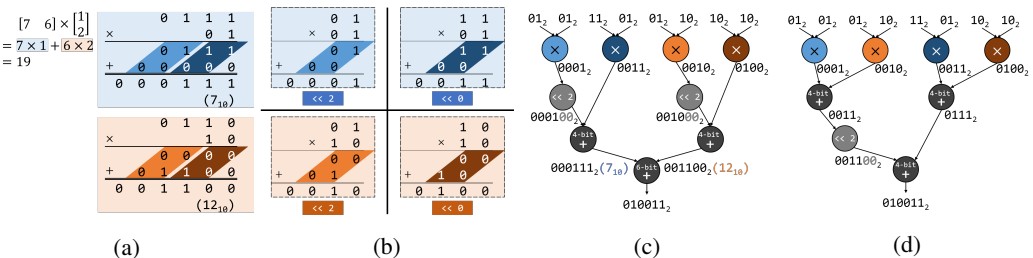

(a)      (b)      (c)      (d)

Figure 3: Example of matrix multiplication

To reduce the power overhead of a decomposable MAC, we first observe the behavior of matrix multiplication. Fig. 3 illustrates an example of matrix multiplication with unsigned integers for simplicity (the extension of signed numbers will be discussed in Section A.1). Suppose that the inputs are matrices $\boldsymbol{A} = [7 \quad 6]$ and $\boldsymbol{B} = [1 \quad 2]^T$, and the result matrix of $\boldsymbol{A} \times \boldsymbol{B}$ will be $[7 \times 1 + 6 \times 2] = [19]$. If $\boldsymbol{A}$ and $\boldsymbol{B}$ are represented with 4-bit and 2-bit binaries, respectively, and only 2-bit×2-bit multipliers are provided, the matrix multiplication should be decomposed and composed to obtain the correct result. As shown in Fig. 3a, each 4-bit×2-bit multiplication is decomposed into two 2-bit×2-bit multiplications. Therefore, this computation needs four 2-bit×2-bit multiplications and two of them have to be shifted left with 2 bits (see Fig. 3b). If we follow a straightforward method presented by Sharma et al. (2018) to obtain "meaningful" intermediate values during the computation, i.e., obtain the result of $7 \times 1$ and then the result of $6 \times 2$, four multiplications, two shift operations and three additions are necessary, as shown in Fig. 3c. This indicates that four 2-bit×2-bit multipliers, two 4-bit adders and one 6-bit adder will be utilized for the computation.

Fig. 3d shows the decomposition and composition flow based on our idea: if we only care the final result but not the intermediate values, the partial results which have the same shift distances can be composed together before the shift operation. Afterwards, this matrix multiplication needs four multiplications, three additions and only one shift operation. Thus, four 2-bit×2-bit multipliers and three 4-bit adders will be utilized.

### 2.2 CONTRIBUTIONS OF OUR PROPOSED DECOMPOSITION STRATEGY

The contributions and advantages of this proposed work are twofold:

- We propose a decomposition strategy for low-power decomposable multiplication-accumulation operations. Based on our decomposition strategy, longer bit-width multiplication can be departed and composed with the results of shorter bit-width MACs. Therefore, compared with a non-decomposable multiplier/MAC, when a DNN has its activations and weights being quantized into variable bit-widths among different layers, our decomposable MAC can generate the matrix multiplication results with less energy consumption.

- According to our proposed decomposition strategy, the partial results which have the same shift distance are composed together before shifted and added with other partial results.

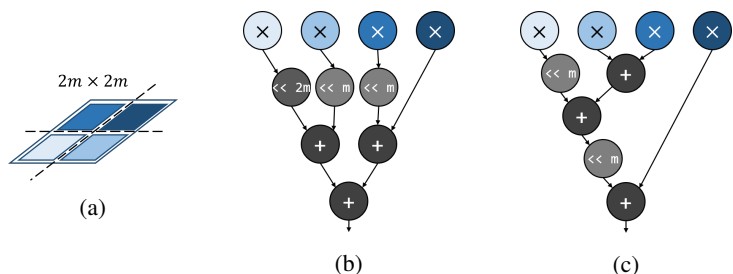

Figure 4: $2m$-bit$\times 2m$-bit multiplication

Instead of designing a decomposable multiplier to construct a systolic array by straightforward decomposition methods, we efficiently utilize the shorter bit-width MACs within a systolic array. Hence, our decomposable MAC has less power overhead of connecting signals among multiplication blocks/modules physically.

# 3 DETAILS OF OUR PROPOSED DECOMPOSABLE MAC

After given an illustrative example for the major idea of our decomposition strategy, we will introduce the detailed concept and architecture of our proposed decomposable MAC in this section.

## 3.1 CONCEPT OF OUR DECOMPOSITION STRATEGY

Our decomposition strategy can be extended to any multiplication with variable bit-widths. As illustrated in Fig. 4a, if the inputs of a multiplication are represented as $2m$-bit integers and $m$-bit$\times m$-bit multipliers are provided, the multiplication can be departed into four $m$-bit$\times m$-bit multiplications. Fig. 4b shows that when the straightforward method presented by Sharma et al. (2018) is applied, it needs two kinds of shifters (one is for left-shift with $2m$ bits and the other is for left-shift with $m$ bits) and three shift operations for the computation. With our decomposition strategy, it needs one shifter and two shift operations as shown in Fig. 4c. For both Fig. 4b and 4c, two $2m$-bit adders and one $3m$-bit adder will be used. If an $m$-bit$\times m$-bit multiplier is provided and the inputs of a multiplication are $l \times m$-bit integers $a$ and $b$, the multiplication result can be obtained by:

$$c = a \times b = (\sum_{i=0}^{l-1} 2^{i \cdot m} a_i)(\sum_{i=0}^{l-1} 2^{i \cdot m} b_i) = \sum_{i=0}^{2l-2}(2^{i \cdot m} \sum_{j=0}^{i} a_j b_{i-j}) \quad (1)$$

where $a_i$ and $b_i$ are $m$-bit integers expressing $a$ and $b$ by $a = \sum_{i=0}^{l-1} 2^{i \cdot m} a_i$ and $b = \sum_{i=0}^{l-1} 2^{i \cdot m} b_i$, respectively. With our decomposition strategy, the multiplication only needs $l^2$ $m$-bit$\times m$-bit multiplications, $2l - 2$ shifts and totally $(4l^2 - 5l + 1) \times m$ 1-bit full adders (FAs), which are much less than $l^2 - 1$ shifts and at least $(l^2 \times \log_2 l + l/2) \times m$ FAs with a straightforward decomposition method.

Furthermore, when our decomposition strategy is applied on matrix multiplication, the number of FAs for the composition of partial results can be reduced significantly. Suppose that the inputs of a matrix multiplication are matrices $A$ and $B$, and the result is matrix $C$, then every element in $C$ can be calculated by: $c_{ij} = \sum_{k=1}^{N} a_k b_k$ where $a_k$ and $b_k$ are the $k_{th}$ element in row $i$ of matrix $A$ and the $k_{th}$ element in column $j$ of matrix $B$, respectively. If the elements in matrices $A$ and $B$ are $2m$-bit integers and the given multiplier can perform $m$-bit$\times m$-bit multiplication, the element $c_{ij}$ can be obtained by departing and composing:

$$
\begin{aligned}
c_{ij} &= \sum_{k=1}^{N}(a_{k1} \times 2^m + a_{k0})(b_{k1} \times 2^m + b_{k0}) \\
&= 2^{2m} \sum_{k=1}^{N}(a_{k1} b_{k1}) + 2^m \sum_{k=1}^{N}(a_{k1} b_{k0} + a_{k0} b_{k1}) + \sum_{k=1}^{N} a_{k0} b_{k0}
\end{aligned}
\quad (2)
$$

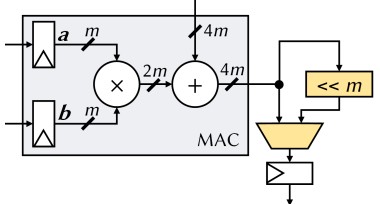

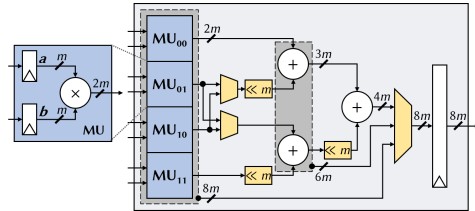

(a) A decomposable MAC based on our proposed decomposition strategy

(b) A configurable multiplier based on the method presented by Sharma et al. (2018)

Figure 5: Decomposable MAC/multiplier for $2m$-bit$\times 2m$-bit multiplication

where $a_{k1}$, $a_{k0}$, $b_{k1}$, and $b_{k0}$ are numbers expressed in $m$ bits and their combination, $a_{k1} \times 2^m + a_{k0}$ and $b_{k1} \times 2^m + b_{k0}$, represent the value of elements $a_k$ and $b_k$, respectively. In addition, the result of multiplication with power of 2 can be obtained by left-shift operation. Furthermore, the shift distance can be a constant with the value being $m$ if Equation 2 is rewritten as:

$$c_{ij} = 2^m[2^m \sum_{k=1}^{N}(a_{k1}b_{k1}) + \sum_{k=1}^{N}(a_{k1}b_{k0} + a_{k0}b_{k1})] + \sum_{k=1}^{N} a_{k0}b_{k0} \quad (3)$$

When the elements in matrices $A$ and $B$ are represented as $l \times m$-bit integers, the elements in result matrix $C$ can be obtained by replacing the expressions of $a_k$ and $b_k$ in Equation 2 as appropriate polynomials in form of $\sum_{k=0}^{l-1} d_k 2^{km}$, where $d_k$ is a $m$-bit integer. In this condition, a straightforward method needs $Nl^2 - N$ shifts, but our decomposition strategy still needs only $2l - 2$ shifts. To perform the matrix multiplication with appropriate hardware, we describe the detailed architecture in the following subsection.

## 3.2 ARCHITECTURE OF OUR PROPOSED DECOMPOSABLE MAC

According to our decomposition strategy, a decomposable MAC can be constructed by connecting the output of an $m$-bit$\times m$-bit MAC to a multiplexer (MUX) for the selection between shifted and non-shifted results as illustrated in Fig. 5a.

Table 1 lists the number of FAs and 1-bit 2-to-1 MUXs of two multipliers and one MAC, including a non-decomposable $2m$-bit$\times 2m$-bit multiplier, a straightforward decomposable multiplier and the MAC based on our decomposition strategy. Suppose that these multipliers use array multipliers for the multiplication and the decomposable ones use ripple-carry adders for the partial product combination (the applied multipliers and adders can be any kind of existing multipliers and adders, and we choose array multipliers and ripple-carry adders here for simplicity). In this condition, the non-decomposable multiplier contains $2m(2m - 1) = 4m^2 - 2m$ FAs.

Based on a straightforward method, a decomposable multiplier requires enough multiplication units in their design, e.g., four $m$-bit$\times m$-bit multipliers are necessary for a $2m$-bit$\times 2m$-bit multiplication. In addition, the decomposable multiplier needs to select the required combination of the results (four $m$-bit$\times m$-bit, two $m$-bit$\times 2m$-bit, two $2m$-bit$\times m$-bit, or one $2m$-bit$\times 2m$-bit multiplication

Table 1: Comparisons of required FAs and MUXs

| | #FAs for multiplication | #FAs for composition | #MUXs |
|---|---|---|---|
| Non-decomposition (one $2m \times 2m$ multiplier) | $4m^2 - 2m$ | 0 | 0 |
| Straightforward decomposition method (four $m \times m$ multipliers) | $4m^2 - 4m$ | $7m$ | $42m$ |
| Our decomposition strategy (one $m \times m$ MAC) | $m^2 - m$ | $4m$ | $4m$ |

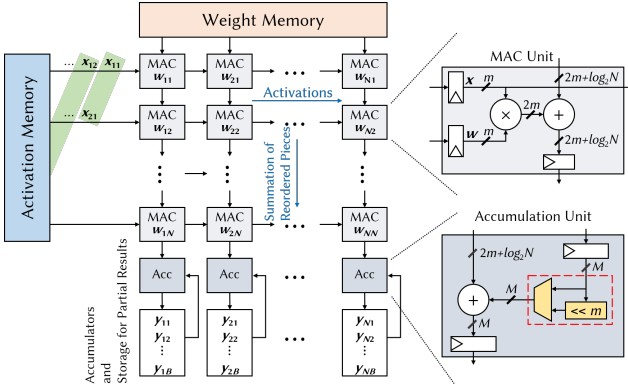

Figure 6: Architecture of our systolic array of decomposable MACs

results), and then add them together. Therefore, $7m$ FAs and $42m$ MUXs are required for the composition of a straightforward decomposable multiplier.

However, the decomposable MAC based on our decomposition strategy can use a single $m$-bit$\times m$-bit multiplication unit and a $4m$-bit adder to achieve scalability, as illustrated in Fig. 5. The number of required MUXs can be $4m$ because our decomposable MAC only needs to select the result with correct shift distance from the accumulation result. Thus, the number of required FAs and MUXs for composition based on our decomposition strategy can be less than those required by the straightforward decomposable multiplier. Because the longest path in our decomposable MAC is shorter, the delay of our decomposable MAC can be less than both delays of the non-decomposable multiplier and the straightforward decomposable multiplier. Accordingly, the energy consumption of our decomposable MAC can also be less.

Our decomposition strategy can be applied on a systolic array (Kung & Leiserson, 1979; Kung, 1982) of MACs to construct a more energy-efficient MAC for DNN computation. Fig. 6 illustrates the block diagram of our decomposable MAC, which consists of a systolic array of an $N \times N$ grid of shorter bit-width MACs and $N$ accumulation units. Each MAC unit can perform $m$-bit$\times m$-bit multiplication and $(2m + \log_2 N)$-bit accumulation, where $m$ is the basic length for the decomposable multiplication and $2m + \log_2 N$ is the minimum length for representing the maximum accumulation result of $N$ MACs within the same column of the systolic array. In addition, each accumulation unit contains an $M$-bit accumulator and a MUX for the selection between shifted and non-shifted partial results, where $M$ is the minimum length for the representation of elements in a result matrix.

In our proposed decomposable MAC, a 2-dimensional systolic array of shorter bit-width MACs is employed to calculate the partial results of matrix multiplication. By connecting multiple MACs as the processing elements in the systolic array, data can be passed and computed through more processing elements after I/O operation. The weights are pre-loaded to the registers of corresponding MACs before the matrix multiplication execution. As shown in Fig. 6, the activations of each layer is propagated from left to right and the partial results of matrix multiplication are accumulated from top to bottom cycle by cycle.

After all the partial results with common shift distance are accumulated together, these pieces should be shifted and composed. As illustrated in Fig. 6, we connect the outputs of the systolic array to MUXs for selection between shifted and non-shifted partial results. Therefore, according to our decomposition strategy, reordering the addition sequence of partial results in matrix multiplication with common shift distances not only reduce the number of shift operations and the required MUXs significantly, but also reduce the connecting signals among multiplication blocks/modules physically.

## 4  EXPERIMENTAL RESULTS

In this section, we compare a systolic array of MACs based on our decomposition strategy to systolic arrays of non-decomposable MACs and straightforward decomposable MACs. In the experiments,

we implement the systolic arrays of MACs in Verilog and the MACs are synthesized by *Synopsys Design Compiler* (DC) with Synopsys SAED EDK 28nm standard cell library. Moreover, the basic arithmetic components, including adders and multipliers, are provided by DesignWare.

### 4.1 BASIC COMPARISONS WITH DIFFERENT SYSTOLIC ARRAY LENGTH

Table 2: Latency, area and energy consumption of a systolic array of MACs

|  | $Latency(ns)$ | $Area(\mu m^2)$ | $Energy(\mathrm{p}J)$ |
|---|---|---|---|
| Non-decomposable | 2.00 | 311679.56 | 54.144 |
| Straightforward decomposable | 2.72 | 601602.42 | 110.989 |
| Our decomposable | 1.16 | 516512.40 | 37.764 |

To evaluate the latency, area and energy consumption of a non-decomposable MAC, a straightforward decomposable MAC and our decomposable MAC, we implement these systolic arrays in Verilog. Three different size of systolic arrays are implemented, including a 16×16 grid of MACs, a 32×32 grid of MACs and 64×64 grid of MACs. To align the capability of performing matrix multiplication in each cycle as the systolic array of straightforward decomposable MACs does, the size of our systolic array is adjusted to be 16×64, 32×128, and 64×256. Within these systolic arrays, the width of accumulators/adders of MACs have been adjusted by considering the corresponding maximum length of accumulation in the systolic array. In addition, all of these three systolic arrays of MACs have ability to compute correct result of multiplication with its input bit-width being 4-bit × 4-bit, 8-bit × 4-bit, and 8-bit × 8-bit.

Table 2 shows the synthesized results of three systolic arrays with a 16×16 grid of non-decomposable MACs, a 16×16 grid of straightforward decomposable MACs, and a 16×64 grid of MACs based on our proposed decomposition strategy. The first column is the type of MACs utilized within the systolic array. The second column shows the latency of each type of MAC. Columns three and four show the area and energy consumption of the whole systolic array of MACs. To accelerate the operations of matrix multiplication, a straightforward decomposable MAC is designed to perform as one MAC with an 8-bit×8-bit multiplier, two parallel MACs with 8-bit×4-bit multipliers, or four parallel MACs with 4-bit×4-bit multipliers. Hence, additional components are necessary for the construction of a straightforward decomposable MAC (as mentioned in Section 3.2), and thus the whole area and energy consumption of the systolic array of straightforward decomposable MACs are larger than those of the systolic array of non-decomposable MACs. To reduce the energy consumption of matrix multiplication, our decomposition strategy reuses the connections within the systolic array to compose partial results of multiplication for the final correct results. Therefore, the input bit-width of our MAC unit just needs to be 4-bit×4-bit, which significantly reduce the area and energy consumption of decomposable MACs.

In Fig. 7, to compare the synthesized results more clearly, we record their energy consumption and performance in three different operation conditions, including multiplication with all the bit-width of activations and weights being 4×4, 8×4, and 8×8, respectively. As shown in Fig. 7a, the energy consumption of the systolic array with our proposed decomposition strategy is much less than that applied a straightforward decomposition method. In Fig. 7b, the performance of both

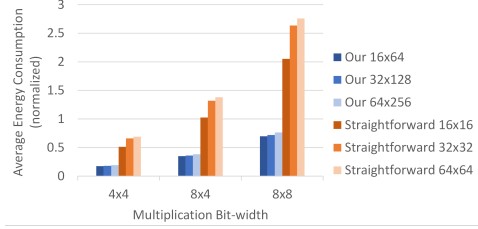

(a) Normalized average energy consumption

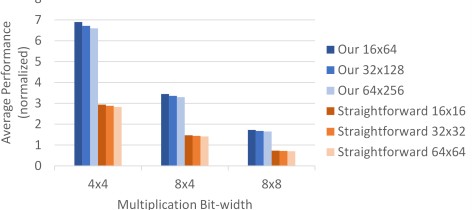

(b) Normalized performance

Figure 7: Normalized average energy consumption and performance of the systolic arrays of decomposable MACs (normalized to those of the systolic array of non-decomposable MACs)

systolic arrays of decomposable MACs are higher than that of non-decomposable MACs when operating multiplication with all the bit-width being 4×4 and 8×4. This is because both arrays of decomposable MACs can generate four times 4-bit×4-bit multiplication results and double 8×4 multiplication results compared with an array of non-decomposable MACs. However, when operating 8-bit×8-bit multiplication, the performance of the systolic array of straightforward decomposable MACs is lower than the non-decomposable one. Due to the connecting signals among the multiplication units of a straightforward decomposable multiplier, its latency will be longer than a non-decomposable multiplier and cause lower throughput when operating as one MAC with an 8-bit×8-bit multiplier. The latency of the MAC based on our proposed decomposition strategy is shorter than the non-decomposable one, and thus the performance of our systolic array can still be higher than the non-decomposable one when operating 8-bit×8-bit multiplication.

## 4.2 ENERGY EFFICIENCY WHEN OPERATING MATRIX MULTIPLICATION OF DEEP NEURAL NETWORKS

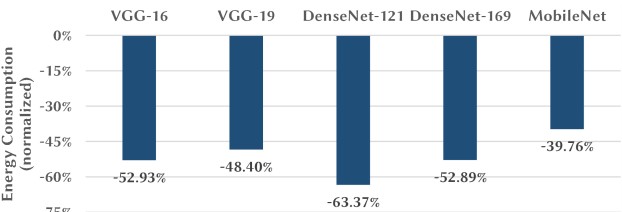

Figure 8: Energy reduction of the systolic array of MACs applied our decomposition strategy (normalized to the energy consumption of the systolic array of non-decomposable MACs)

To evaluate the energy efficiency of the systolic array of MACs applying our decomposition strategy, we perform matrix multiplication in deep learning applications according to the bit-width requirement mentioned in Section 1 with the synthesized systolic arrays shown in Table 2.

Fig. 8 shows the energy consumption of the systolic array of MACs based on our decomposition strategy compared with a systolic array of non-decomposable MACs. When a DNN has its activations and weights being quantized into shorter bit-width, the systolic array of non-decomposable MACs still operate 8-bit×8-bit multiplication within its multiplication units, and thus it cannot efficiently obtain the benefit energy saving from the quantization. However, due to applying shorter bit-width MACs for the decomposition and MUXs for the selection between shifted and non-shifted accumulation results for the composition, our decomposable MAC has significant energy reduction when the required bit-width of multiplication is shorter.

## 5 CONCLUSION

In this paper, we propose a decomposition strategy for the energy efficiency of quantized DNN computation. According to our proposed strategy, longer bit-width multiplication can be departed and composed with partial results of shorter bit-width MACs. Therefore, compared with a non-decomposable MAC, our decomposable MAC can generate matrix multiplication results with less energy consumption, when a DNN has its activations and weights being quantized into variable bit-widths among different layers. In addition, we efficiently utilize the MACs within a systolic array for the composition of partial results. Hence, a systolic array of MACs based on our decomposition strategy has less power overhead than a systolic array of straightforward decomposable MACs.

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

## A  APPENDIX

### A.1  EXTENSION TO SIGNED MULTIPLICATION

Our proposed decomposition strategy can be applied on signed multiplication by replacing the m-bit×m-bit unsigned multiplier into a multiplier which can perform combined unsigned/signed multiplication. For example, according to the two's complement multiplication developed by Baugh & Wooley (1973), a 4-bit×4-bit signed multiplication can be illustrated as Fig. 9a. The correct result of signed multiplication can be obtained by inverting some of the product terms and inserting a one to the left of the first partial product term. To support decomposable signed multiplication, flag signals $S_a$ for the determination of the most significant bit (MSB) of signed integer $a$, $S_b$ for the

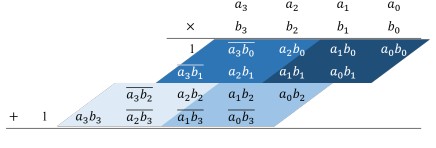

(a) A 4-bit×4-bit signed multiplication

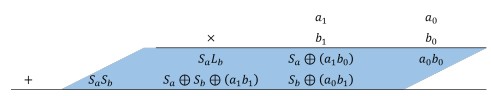

(b) A 2-bit×2-bit multiplier for combined unsigned/signed multiplication

Figure 9: A signed multiplication with decomposable multipliers

determination of the MSB of signed integer $b$, and $L_b$ for the determination of the least significant bit of $b$ should be provided in each 2-bit×2-bit multiplier. When the partial product terms of a 2-bit×2-bit multiplier which replaces the unsigned multiplier is the same as those shown in Fig. 9b, the decomposable multiplier can support signed multiplication.

