# OpenReview forum: "Efficient Systolic Array Based on Decomposable MAC for Quantized Deep Neural Networks"
_ICLR.cc/2020/Conference — Reject_

### Official Review · AnonReviewer2 · 2019-10-24
**Official Blind Review #2**

**Rating:** 1

**Review:**

This paper proposes to shorten the shift-addition operations in the straightforward configurable MACs (Sharma et al., 2018), to an addition-shift style. The authors claim that the new design is able to lower the energy consumption in the matrix multiplication. In the experimental analysis, the authors demonstrate the effectiveness of the proposed method.

This paper should be rejected since it proposes exactly the same architecture with the following published work:
BitBlade: Area and Energy-Efficient Precision-Scalable Neural Network Accelerator with Bitwise Summation

Also, the authors do not provide a valid approach for the auto-selection of quantization bits, which is more significant in my opinion.

**Experience Assessment:**

I have read many papers in this area.

**Review Assessment: Checking Correctness Of Derivations And Theory:**

I assessed the sensibility of the derivations and theory.

**Review Assessment: Checking Correctness Of Experiments:**

I assessed the sensibility of the experiments.

**Review Assessment: Thoroughness In Paper Reading:**

I made a quick assessment of this paper.

---

### Official Review · AnonReviewer1 · 2019-10-27
**Official Blind Review #1**

**Rating:** 3

**Review:**

This paper presents a decomposable MAC unit for low-power computation for matrix-multiplication operation for neural networks.

Although I believe I understood the idea of this technique, I strongly believe this paper should be submitted to an architecture of design automation conference instead of ICLR. I am also not in the position to assess the experiments, which were conducted with synthesis of Synopsis 28nm library. The paper discussed systolic array, MAC in detail, without too many algorithmic elements inside, therefore may be a paper not toward the audience of this conference.

**Experience Assessment:**

I have published one or two papers in this area.

**Review Assessment: Checking Correctness Of Derivations And Theory:**

I assessed the sensibility of the derivations and theory.

**Review Assessment: Checking Correctness Of Experiments:**

I assessed the sensibility of the experiments.

**Review Assessment: Thoroughness In Paper Reading:**

I read the paper at least twice and used my best judgement in assessing the paper.

---

### Official Review · AnonReviewer3 · 2019-11-01
**Official Blind Review #3**

**Rating:** 3

**Review:**

Summary:

This paper proposes a novel decomposition strategy for matrix multiplication to aim for less energy consumption. They demonstrate that energy consumption is reduced on several CNNs. It is an interesting work but it is not sure that there is a difference in contribution when compared to previous work.

The main argument to impact to the score:

    1. A similar idea is addressed in [1]. It is highly recommended to show the difference in contribution.
    2. In Figure 7,  the performance and energy consumption of systolic arrays of decomposable MAC  is shown. However, the only energy consumption is shown in section 4.2 when the method is applied to DNNs. It is better to show both performance and energy consumption.
    3. It is inappropriate to quantize VGG16, VGG19, DenseNet-121, DenseNet-169 and MobileNet based on CIFAR10 dataset since all models are designed for ImageNet dataset. It is appropriate to quantize these models on ImageNet dataset.


[1]     Sungju Ryu, Hyungjun Kim, Wooseok Yi and Jae-Joon Kim. BitBlade: Area and Energy-Efficient Precision-Scalable Neural Network Accelerator with Bitwise Summation. DAC. 2019.

Minor comments not to impact the score:
    1. "Very Deep Convolutional Networks for Large-Scale Image Recognition" has been accepted by ICLR 2014. It is better not to use arXiv preprint on citations unless there is a reason.
    2. It is recommended to put citations on CIFAR-10 dataset.

**Experience Assessment:**

I do not know much about this area.

**Review Assessment: Checking Correctness Of Derivations And Theory:**

I assessed the sensibility of the derivations and theory.

**Review Assessment: Checking Correctness Of Experiments:**

I carefully checked the experiments.

**Review Assessment: Thoroughness In Paper Reading:**

I read the paper at least twice and used my best judgement in assessing the paper.

---

### Public Comment · ~gang_li1 · 2019-10-24
**Overlap with previous work**

This paper proposes decomposable MAC and a systolic array-based accelerator for varible bitwidth of weights/activations in CNN. The core insight is that the partial results of a multiplication which have the same shift distance can be composed together before shifted so as to reduce area and energy consumption.

However, exactly the same idea has been seen in previous work BitBlade [1], which was also an incremental work based on Bit Fusion [2]. The difference is that BitBlade adopted a 2D-mesh architecture instead of systolic array.

I think the authors should cite BitBlade in the paper and reconsider the contributions.

Thanks.



[1] Sungju Ryu, Hyungjun Kim, Wooseok Yi, Jae-Joon Kim. BitBlade: Area and Energy-Efficient Precision-Scalable Neural Network Accelerator with Bitwise Summation. DAC'2019
https://dl.acm.org/citation.cfm?id=3317784

[2] Hardik Sharma et al. Bit Fusion: Bit-Level Dynamically Composable Architecture for Accelerating Deep Neural Networks. ISCA'2018

---

### Decision · Program_Chairs · 2019-12-19

**Decision:**

Reject

**Comment:**

This paper presents an energy-efficient architecture for quantized deep neural networks based on decomposable multiplication using MACs. Although the proposed approach is shown to be somehow effective, two reviewers pointed out that the very similar idea was already proposed in the previous work, BitBlade [1]. As the authors did not submit a rebuttal to defend this critical point, I’d like to recommend rejection. I recommend authors to discuss and clarify the difference from [1] in the future version of the paper.

[1] Sungju Ryu, Hyungjun Kim, Wooseok Yi, Jae-Joon Kim. BitBlade: Area and Energy-Efficient Precision-Scalable Neural Network Accelerator with Bitwise Summation. DAC'2019